# Antimicrobial Peptides Epinecidin-1 and Beta-Defesin-3 Are Effective against a Broad Spectrum of Antibiotic-Resistant Bacterial Isolates and Increase Survival Rate in Experimental Sepsis

**DOI:** 10.3390/antibiotics11010076

**Published:** 2022-01-09

**Authors:** Albert Bolatchiev

**Affiliations:** Department of Clinical Pharmacology, Stavropol State Medical University, 355000 Stavropol, Russia; bolatalbert@gmail.com

**Keywords:** antibiotic resistance, antimicrobial peptides, defensin, epinecidin, carbapenem-resistance, methicillin-resistance

## Abstract

The antimicrobial peptides human Beta-defensin-3 (hBD-3) and Epinecidin-1 (Epi-1; by *Epinephelus coioides*) could be a promising tool to develop novel antibacterials to combat antibiotic resistance. The antibacterial activity of Epi-1 + vancomycin against methicillin-resistant *Staphylococcus aureus* (22 isolates) and Epi-1 + hBD-3 against carbapenem-resistant isolates of *Klebsiella pneumoniae* (*n* = 23)*,* *Klebsiella aerogenes* (*n* = 17)*, Acinetobacter baumannii* (*n* = 9), and *Pseudomonas aeruginosa* (*n* = 13) was studied in vitro. To evaluate the in vivo efficacy of hBD-3 and Epi-1, ICR (CD-1) mice were injected intraperitoneally with a lethal dose of *K. pneumoniae* or *P. aeruginosa*. The animals received a single injection of either sterile saline, hBD-3 monotherapy, meropenem monotherapy, hBD-3 + meropenem, or hBD-3 + Epi-1. Studied peptides showed antibacterial activity in vitro against all studied clinical isolates in a concentration of 2 to 32 mg/L. In both experimental models of murine sepsis, an increase in survival rate was seen with hBD-3 monotherapy, hBD-3 + meropenem, and hBD-3 + Epi-1. For *K.* *pneumoniae*-sepsis, hBD-3 was shown to be a promising option in overcoming the resistance of *Klebsiella* spp. to carbapenems, though more research is needed. In the *P. aeruginosa*-sepsis model, the addition of Epi-1 to hBD-3 was found to have a slightly reduced mortality rate compared to hBD-3 monotherapy.

## 1. Introduction

The global spread of antimicrobial resistance within bacteria is a serious threat, causing many antibiotics to lose their effectiveness [1,2]. According to the United States (US) Center for Disease Control and Prevention’s (CDC) latest report, more than 2.8 million antibiotic-resistant infections occur in the US each year, and more than 35,000 people die as a result [3]. According to this report, one of the most urgent threats is carbapenem-resistant *Enterobacterales*: these bacteria account for more than 1100 deaths in the US annually, and cause more than $130 million in financial losses every year. Multi-drug resistant *Pseudomonas aeruginosa* has been designated a serious threat, causing more than 2700 annual deaths and resulting in a $767 million financial burden on the US healthcare system. Overall, antibiotic-resistant infections cause more than 700,000 deaths worldwide every year, and it is projected that, by the middle of the 21st century, this figure will exceed the death rate of cancer and result in more than 10 million deaths a year [2]. In this regard, there is an urgent need to discover new treatment options for infections caused by resistant microorganisms.

One of the strategies to combat antibiotic resistance is to use antimicrobial peptides (AMPs), which are an essential component of the innate immune system and protect against bacteria, viruses, fungi, parasites, and tumor cells [4]. While conventional antibiotics act on one specific target, AMPs simultaneously affect membranes and intracellular targets of a bacterial cell, thereby providing high activity against resistant bacteria [5,6]. AMPs are mainly hydrophobic peptides with a positive charge. Their effectiveness is dependent on charge neutralization and penetration through bacterial membranes, producing bacterial cell death and reducing the likelihood for the development of bacterial resistance [7]. AMPs are safer and less likely to have toxic side effects compared to conventional antibiotics [8].

Previously, we investigated the in vitro activity of human AMPs from the defensins class against bacterial strains of *Staphylococcus aureus* and *Escherichia coli,* collected from hospitalized patients [9]. Research has shown the most effective human defensins to be human beta-defensin-3 (hBD-3), which is produced by epithelial cells. We showed that human hBD-3 is highly active against methicillin-resistant *S. aureus* (MRSA) and carbapenem-resistant *E. coli*. In addition, hBD-3 showed a pronounced synergistic effect when used together with rifampicin (against MRSA) and amikacin (against *E. coli*). Several studies have shown various AMPs to exhibit synergistic effects in vitro when combined with carbapenems [10]. However, more research is needed to evaluate the effects of AMPs and carbapenems in experimental in vivo models.

Epinecidin-1 (Epi-1) is an AMP derived from the orange-spotted grouper (*Epinephelus coioides*) and was discovered in 2005 [11]. Epi-1 has a wide spectrum of antimicrobial activity against Gram-positive and Gram-negative bacteria and viruses, in addition to an antitumor effect, a wound healing effect, and an immune response regulator [12,13]. Despite numerous studies devoted to studying the properties of Epi-1, there is insufficient data documenting its effectiveness against resistant Gram-negative infections in vivo. The activity of Epi-1 in combination with any human AMPs has not been previously studied. In addition, the antibacterial effect of the combination of Epi-1 and vancomycin against MRSA has not been previously evaluated. In one study, the effectiveness of the combined use of Epi-1 + vancomycin was analyzed in experimental MRSA infection in pigs [14]; however, the classical in vitro checkerboard assay for assessing the mutual antibacterial effect of two substances was not used here [9,15,16,17,18].

The aim of this study was to evaluate the antimicrobial activity of: (i) Epi-1 + vancomycin against MRSA in vitro; (ii) Epi-1 + hBD-3 against carbapenem-resistant isolates of *Klebsiella pneumoniae, Klebsiella aerogenes, Acinetobacter baumannii,* and *P. aeruginosa* in vitro; (iii) Epi-1 + hBD-3 in the experimental model of *K. pneumoniae*-induced sepsis.

## 2. Results

Before describing the results, the physicochemical characteristics (Table 1) and three-dimensional structures (Figure 1) of the studied AMPs are presented below.

### 2.1. Epi-1 Is Effective against MRSA and Carbapenem-Resistant Gram-Negative Bacteria In Vitro

In the first part of this study, the antibacterial activity of Epi-1 with vancomycin against MRSA and Epi-1 with hBD-3 against carbapenem-resistant isolates of *K. pneumoniae* (*n* = 23), *K. aerogenes* (*n* = 17), *P. aeruginosa* (*n* = 13), and *A. baumannii* (*n* = 9) was studied. Minimum inhibitory concentration (MIC) and fractional inhibitory concentration index (FICI) values were determined (presented as median and first and third quartile in parentheses). 

The MIC of Epi-1 against MRSA was 16 (14–16) mg/L. MIC of vancomycin was 2 (2–2) mg/L. The combination of Epi-1 with vancomycin showed a synergistic effect against 10 MRSA isolates. In the other 12 cases, Epi-1 and vancomycin had no effect on the activity of each other. The median fractional inhibitory concentration index (FICI) value was 0.75 (0.5–0.75) (Table 2).

The MIC of hBD-3 for *K. pneumoniae* was 4 (4–8) mg/L; for *K. aerogenes*, 4 (2–4) mg/L; for *P. aeruginosa*, 16 (8–16) mg/L; and for *A. baumannii*, 8 (4–12) mg/L (Table 3, Table 4, Table 5 and Table 6). 

Epi-1 showed similar performance of MICs with a value of 8 (8–12) mg/L for *K. pneumoniae*, 8 (4–16) mg/L for *K. aerogenes*, 4 (4–8) mg/L for *P. aeruginosa* and 16 (12–32) mg/L for *A. baumannii* (Table 3, Table 4, Table 5 and Table 6).

The combination of hBD-3 and Epi-1, in most cases, did not have a synergistic effect—these AMPs generally do not affect each other’s effectiveness in any way, although for some isolates the combined use of hBD-3 and Epi-1 produced a synergistic effect. The median FICI value for *K. pneumoniae* was 1 (0.69–1.1); for *K. aerogenes*, 0.75 (0.5–0.75); for *P. aeruginosa*, 0.75 (0.5–1); and for *A. baumannii*, 0.5 (0.5–0.75) (Table 3, Table 4, Table 5 and Table 6). 

### 2.2. hBD-3 and Epi-1 Decrease Mortality in Experimental Sepsis

In the second part of this study, the efficacy of hBD-3 and Epi-1 peptides in a murine sepsis model were studied. In the CRKP_1 (Table 3) and CRPA_3 (Table 5) isolates, the combination of hBD-3 + Epi-1 did not show a synergistic effect in the in vitro experiments, and thus was selected to infect the animals.

In the model of *K. pneumoniae*-sepsis, the survival rate after 24 h for the control group was 26.7%; for hBD-3, 86.7%; for meropenem, 53.3%; for hBD-3 + meropenem, 100%; and for hBD-3 + Epi- 1, 93.3%. By the end of the observation period (120 h after infection), the percentage of surviving mice in each group was as follows (exact *p*-values are indicated in parentheses in comparison with the control): control, 0%; hBD-3, 63.3% (*p* = 0.000000003); meropenem, 0% (*p* = 0.036); hBD-3 + meropenem, 83.7% (*p* = 0.0000000000001); and hBD-3 + Epi-1, 70% (*p* = 0.00000000009) (Figure 2). The statistically significant difference between the control group and the meropenem treatment group is clinically irrelevant, as in both cases all animals died—meropenem only slightly delayed fatal outcome (Figure 2). Statistical analysis also revealed a significant difference between the hBD-3 and hBD-3 + meropenem groups (*p* = 0.03). There was no significant difference between hBD-3 and hBD-3 + Epi-1, nor between hBD-3 + Epi-1 and hBD-3 + meropenem. Therefore, the combination of hBD-3 + meropenem and hBD-3 + Epi-1 had the greatest effect on survival in the *K. pneumoniae*-induced sepsis murine model.

When analyzing the results of the experiment using the *P. aeruginosa* isolate, similar data were obtained (Figure 3). At 24 h post-infection, the survival rate was 10% in the control group; for hBD-3, 93.3%; for meropenem, 20%; for hBD-3 + meropenem, 96.7%; and for hBD-3 + Epi-1, 100%. After 5 days, the survival rate between the control and meropenem groups did not differ significantly and was 0%. The survival rate in the hBD-3 group was 66% (significant difference from the control; *p* = 0.000000000004); in the hBD-3 + meropenem group, 63.3% (*p* = 0.000000000001); and in the hBD-3 + Epi-1 group, 86.7% (*p* = 0.00000000000003). There was no significant difference between the hBD-3 and hBD-3 + meropenem groups. The combination of hBD-3 + Epi-1 differed significantly from the hBD-3 group (*p* = 0.046), but did not differ significantly from the hBD-3 + meropenem group. Accordingly, we can assume that the effectiveness of administering hBD-3, hBD-3 + meropenem, or hBD-3 + Epi-1 in *P. aeruginosa*-induced sepsis is approximately the same.

## 3. Discussion

In this study, the antimicrobial activity of the Epi-1 in combination with vancomycin against 22 clinical isolates of methicillin-resistant *S. aureus* was, for the first time, evaluated. 

Different pharmacological effects of Epi-1 have been previously shown in vitro and in experimental animal models [13]. We showed that Epi-1 is effective against a wide spectrum of MRSA clinical isolates, with an MIC range from 16 to 32 mg/L. In most previous studies, the range of MIC values for various *S. aureus* strains (including MRSA) ranged from 6 to 50 mg/L. We also demonstrated the synergistic effect of Epi-1 + vancomycin against some MRSA isolates. In one study, the antistaphylococcal activity of Epi-1 was investigated when combined with conventional antibiotics (kanamycin or streptomycin): a synergistic effect was shown with FICI = 0.4 [19]—the combined use of Epi-1 with these antibiotics resulted in a threefold decrease in MIC values. The authors suggested that the cause of synergy may be a different mechanism of action than that of aminoglycosides (namely, the effect on the translation of proteins in microbial cells) [20] and AMPs (membrane permeabilization) [7]. In addition, in this study, the authors obtained various modifications of truncated Epi-1, which also showed pronounced antibacterial activity. There are no other publications on the combined effect of Epi1-1 with other antimicrobial drugs/peptides.

Vancomycin is a glycopeptide antibiotic. It binds to D-alanyl D-alanine, which inhibits peptidoglycan synthase (glucosyltransferase), and the P-phospholipid carrier, thus preventing the synthesis and polymerization of N-acetylmuramic acid and N-acetylglucosamine within the peptidoglycan layer. This, in turn, weakens the bacterial cell wall, and leakage of intracellular components occurs, which causes the death of bacteria [21]. Therefore, AMPs and vancomycin have different mechanisms of action, which can presumably lead to a synergistic effect. Previously, several studies have shown that vancomycin and various natural and newly designed AMPs can exhibit a synergistic effect against Gram-positive bacteria [22,23,24,25]. 

This study was the first investigation of the combined effect of Epi-1 with human AMP (namely, hBD-3) against broad spectrum of carbapenem-resistant strains of Gram-negative bacteria *K. pneumoniae* (*n* = 23), *K. aerogenes* (*n* = 17), *P. aeruginosa* (*n* = 13), and *A. baumannii* (*n* = 9). 

Interestingly, the effectiveness of Epi-1 has not been previously studied in relation to these bacteria. In one of the first works on the Epi-1, it was shown that its MIC was 100 mg/L against *Klebsiella oxytoca* [26]. We showed that both peptides exhibited antibacterial effects in the concentration range of 2 to 32 mg/L. In more than half of the cases for *A. baumannii*, in almost half of *P. aeruginosa* isolates, and in almost one third of cases for *Klebsiella* spp., the Epi-1 + hBD-3 combination showed synergy. We chose to study hBD-3, as we have previously investigated its activity in combination with amikacin against *E. coli*, where classical synergism has also been demonstrated [9]. These results provide new data that can be used in developing new treatments for multidrug-resistant bacterial infections. 

It remains unclear why in some cases Epi-1 + hBD-3 enhanced each other’s effectiveness, and in some did not. Here it is necessary to conduct research on an even larger number of bacterial strains, as well as to study this effect at the molecular level. It seems that the phenotype of antibiotic resistance may somehow influence this, though many studies have shown that the antibiotic resistance phenotype of bacteria does not matter for the effectiveness of AMPs [27]. Therefore, it can be speculated that this phenomenon (differences of FICI values for the same species of bacteria) can be explained by some surface characteristics of the bacterial cell wall, which is the main target for AMPs. 

From a clinical point of view, it could seem that MIC values in the range of 2–32 mg/L is a hard-to-reach concentration at the site of infection. Therefore, the purpose of future research should be to find and develop any modifications and/or derivatives of Epi-1. However, it is worth noting that Epi-1 was found to be highly effective in combating *Pseudomonas aeruginosa*-induced peritonitis infection in mouse models, without side effects or toxicity—Epi-1 increased survival in mice [28]. In another study it was shown that, in the pyemia model in pigs, Epi-1 protects animals against MRSA-mediated mortality [14]. 

In an experimental model of murine sepsis caused by *K. pneumoniae* or *P. aeruginosa*, we examined, for the first time, the efficacy of hBD-3 monotherapy in comparison with the following therapies: meropenem, hBD-3 + meropenem, and hBD-3 + Epi-1. For these animal studies, isolates were selected in which synergism was not shown during in vitro tests using the combination of hBD-3 and Epi-1. 

In the *K. pneumoniae*-sepsis model, the addition of Epi-1 to hBD-3 does not significantly reduce mortality compared to hBD-3 monotherapy. However, in the *P. aeruginosa*-sepsis model, slight differences were found between the hBD-3 and hBD-3 + Epi-1 groups. We can cautiously assume that, in these mice, Epi-1 somewhat enhanced the effect of hBD-3. Although, at the same time, there was no difference between hBD-3 + meropenem compared to hBD-3 + Epi-1. One of the limitations of our work is that we did not investigate the effectiveness of Epi-1 monotherapy. In an earlier study, using a *P. aeruginosa*-sepsis model, Epi-1 demonstrated reduced mortality in mice due to its antimicrobial and pronounced immunomodulatory effect [28]. Epi-1 enhances the production of immunoglobulin G by activating the Th2-cell response [29], and reduces the level of tumor necrosis factor-alpha by decreasing the levels of endotoxins [28]. Based on these findings, Epi-1 and hBD-3 (and their derivatives) are the promising new therapeutic agents in the fight against antibiotic-resistant infections.

In the *K. pneumoniae*-sepsis model, we found a weak statistically significant difference when comparing hBD-3 monotherapy to hBD-3 + meropenem. It can be speculated that hBD-3 overcomes the resistance of *K. pneumoniae* to carbapenems in vivo, though additional studies are needed to confirm this hypothesis. Another limitation of this study is that we did not evaluate the in vitro combination of meropenem + hBD-3. Previously, several studies have shown that various AMPs demonstrate synergistic effects with beta-lactams [10,30,31,32]. However, we were unable to find studies on the combination of hBD-3 and carbapenems. We have previously reported that hBD-3 + amikacin has a synergistic effect against several *E. coli* strains [9]. In the same study, we demonstrated that the antibiotic resistance phenotype does not impact the effectiveness of AMPs.

The hBD-3 peptide reduced mortality, in both models of sepsis, when given as monotherapy. It can be assumed that this is directly associated with its antimicrobial effect, as well as some of the immunomodulatory effects of beta-defensins: chemoattraction of T-lymphocytes and dendritic cells, interaction with the CCR6 receptor, and increased production of IL-18. In addition, hBD-3 does not show cytotoxic or hemolytic effects at antimicrobial concentrations [33].

## 4. Materials and Methods

### 4.1. Peptides 

In our study, we evaluated two AMPs, hBD-3 and Epi-1, produced by solid-phase peptide synthesis by AtaGenix Laboratories (Wuhan, China, purity ≥ 95%). A few of the physicochemical properties and amino acid sequences of the studied peptides are presented in Table 1. The three-dimensional structure of hBD-3 is presented in Figure 1a [34], and, for the first time, we used the recently published algorithm AlphaFold to model a highly accurate structure of Epi-1, as shown in Figure 1b [35]. The PyMOL Molecular Graphics System, Version 2.5.2 Schrödinger, LLC (New York, NY, USA) was used to generate the images of the AMPs.

### 4.2. Bacterial Isolates 

The bacterial strains used were collected from hospitalized patients in the intensive care units of hospitals within Stavropol, Russia, from January 2021 to May 2021. All bacterial isolates used in this study were kindly provided by Dr. Elena Kunitsyna. The strains were identified, and their antibiotic susceptibility was determined, using the disk diffusion method, as part of a routine microbiological study at the Department of Clinical Microbiology of the Center of Clinical Pharmacology and Pharmacotherapy in Stavropol, Russia [36]. 

The resistance of *S. aureus* (*n* = 22) to cefoxitin (with zone diameter breakpoint <22 mm) was considered as a marker of methicillin resistance. All *S. aureus* strains were susceptible to vancomycin, linezolid, ceftaroline, and ceftobiprole. According to EUCAST breakpoints, *K. pneumoniae* (*n* = 23), *K. aerogenes* (*n* = 17), *P. aeruginosa* (*n* = 13), and *A. baumannii* (*n* = 9) isolates were resistant to all carbapenem antibiotics. 

*Klebsiella* spp. isolates were sensitive only to tigecycline; *P. aeruginosa* strains were sensitive only to ceftazidime/avibactam; and *A. baumannii* isolates were susceptible to tigecycline and polymyxin B (EUCAST breakpoints used). Colistin sensitivity was not tested, as this medication is not included in routine microbiological testing in Russia. 

Antibiotic resistance phenotypes are not reported in this work, as AMPs act on bacteria regardless of their resistance phenotype; known resistance mechanisms do not act on them [27], which were shown previous work [9].

### 4.3. Study of the Minimum Inhibitory Concentrations and Combined Antimicrobial Effect of Epi-1 + Vancomycin and Epi-1 + hBD-3

The standard broth dilution/checkerboard method was used to study the antimicrobial action of AMPs and vancomycin [17], according to EUCAST guidelines [37]. This is the same method used in a previous study [9]. 

Briefly, pure bacterial colonies were cultured using solid nutrient media (mannitol salt agar, BioMedia, St. Petersburg, Russia). From a fresh morning culture, a suspension was prepared in sterile saline, corresponding to a McFarland turbidity standard of 0.5 (equivalent to 1–2 × 10^8^ CFU/mL). The resulting suspension was dissolved in Mueller-Hinton broth (BBL™ Mueller Hinton Broth, Becton, Dickinson and Company, Franklin Lakes, NJ, USA) to obtain an inoculum with an approximate concentration of 5 × 10^5^ CFU/mL. The inoculum (100 μL) was then added to the wells of a sterile 96-well microtiter plate with a U-shaped bottom (Medpolymer, St. Petersburg, Russia). Next, serial two-fold dilutions were performed, and combinations of two test antimicrobial agents were added to the wells—50 μL of each was used. Control wells were also used; one as a sterility control (Mueller Hinton Broth only, no bacteria) and one as a growth control (bacterial inoculum without AMPs/vancomycin). The plates were then incubated in a thermostat at 37 °C. After 18–20 h, the minimum inhibitory concentrations (MIC) were calculated. The MIC value was considered the minimum concentration of AMPs/vancomycin at which there was no visual growth in the corresponding well [17]. The concentration ranges of hBD-3 and Epi-1 investigated were from 0 to 64 mg/L; for vancomycin, 0 to 8 mg/L. 

The antimicrobial effect of the combinations Epi-1 + hBD-3 and Epi-1 + vancomycin was evaluated using the fractional inhibitory concentration index (FICI) [38]: FICI = (A/MIC A) + (B/MIC B). A and B represent the concentrations of two antimicrobial agents in the mixture that inhibit bacterial growth. MIC A and MIC B are the minimum inhibitory concentrations of antimicrobial agents when applied separately. According to the FICI value: (1) FICI *≤* 0.5, synergism of action; (2) 0.5 < FICI < 4, no interaction; (3) FICI > 4, antagonism of action [39]. The final MIC and FICI values were calculated as median values from three independent experiments against each bacterial isolate. 

### 4.4. Study of the Effect of hBD-3 and Epi-1 in an Experimental Model of Sepsis

The animal studies performed were approved by the local ethics committee at the Stavropol State Medical University (minutes of the meeting No. 95 of 02/18/2021) and were carried out in accordance with the Code of Ethics of the World Medical Association (Declaration of Helsinki, EU Directive 2010/63/EU for animal experiments). The animal studies are reported in compliance with the ARRIVE guidelines [40]. Randomization of animals was carried out to generate groups of equal size. Blinding was not possible in this study, as it was performed by one investigator.

Laboratory mice ICR (CD-1) (females, average weight 30 g) were kept in the vivarium of the Stavropol State Medical University. The animals were housed in temperature-controlled rooms with 6 mice per cage on a 12-h light cycle, at 24 °C and 50–60% humidity, with water and food availability *ad libitum*.

The animals were injected with bacterial suspensions using one of the *K. pneumoniae* or *P. aeruginosa* isolates prepared from a fresh morning culture in accordance with the McFarland turbidity standard 15 (which corresponds to an approximate concentration of 4.5 × 10^9^ CFU/mL). To determine the turbidity of the suspension, a DEN-1 densitometer (Biosan, Latvia) was used. The bacterial suspension and AMPs were dissolved in sterile saline and injected intraperitoneally (the total volume of injected fluid did not exceed 250 μL).

A separate experiment was performed for each bacterial strain (*K. pneumoniae* and *P. aeruginosa*). The overall design of the experiments did not differ. At t = 0 min, all animals were infected with the corresponding isolate in the amount of 6.75 × 10^8^ CFU (150 μL). Thirty minutes after infection, the mice were administered a single injection according to groups: Group 1 (control), sterile saline; Group 2, hBD-3; Group 3, meropenem (Pfizer, New York, NY, USA); Group 4, hBD-3 + meropenem; Group 5, hBD-3 + Epi-1. Each group consisted of thirty mice. The survival rate was assessed every 24 h for 5 days. Moribund animals were killed humanely to avoid unnecessary distress. In each group, the dosage of substances used was the same: hBD-3 and Epi-1, 10 mg/kg; and meropenem, 25 mg/kg. Meropenem dosage was chosen in accordance with previous study of H. Van der Weide et al. [41].

### 4.5. Statistical Analysis

MICs and FICIs (medians, first, and third quartile) were calculated using Numbers 11.1 for macOS (Apple Inc., Cupertino, CA, USA). These are presented in the Appendix A. Survival analyses using the Kaplan-Meier method and Log-rank (Mantel-Cox) test were carried out using GraphPad Prism version 9.2.0 for macOS, GraphPad Software, San Diego, CA, USA, www.graphpad.com, accessed on 21 December 2021 (Raw data presented in Appendix A).

## 5. Conclusions

In this study, we demonstrated that the antimicrobial peptides, Epi-1 and hBD-3, have a broad spectrum of activity against drug-resistant bacteria in vitro. We showed that Epi-1 + vancomycin might be a synergistic combination against methicillin resistant *S. aureus*, as well as Epi-1 + hBD-3 against Gram-negative carbapenem-resistant strains. In experimental models of murine sepsis, it was demonstrated that hBD-3 monotherapy, as well as combinations of hBD-3 + meropenem and hBD-3 + Epi-1, increase survival rates. In *K. pneumoniae*-sepsis, hBD-3 could be a promising therapeutic agent for overcoming the resistance of *Klebsiella* spp. to carbapenems, though more research is needed. In the *P. aeruginosa*-sepsis model, the addition of Epi-1 to hBD-3 was found to slightly reduce mortality compared to hBD-3 monotherapy. Thus, peptides hBD-3 and Epi-1 are promising tools in the development of new treatment options for multidrug-resistant infections caused by carbapenem-resistant bacteria.

## Figures and Tables

**Figure 1 antibiotics-11-00076-f001:**
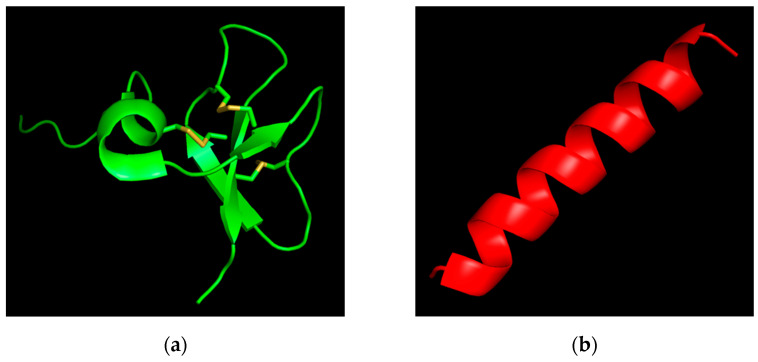
Spatial structure of the studied antimicrobial peptides: (**a**) Human beta-defensin-3 (PDB ID: 1KJ6); (**b**) Epinecidin-1 of the orange-spotted grouper, *Epinephelus coioides* (predicted by AlphaFold).

**Figure 2 antibiotics-11-00076-f002:**
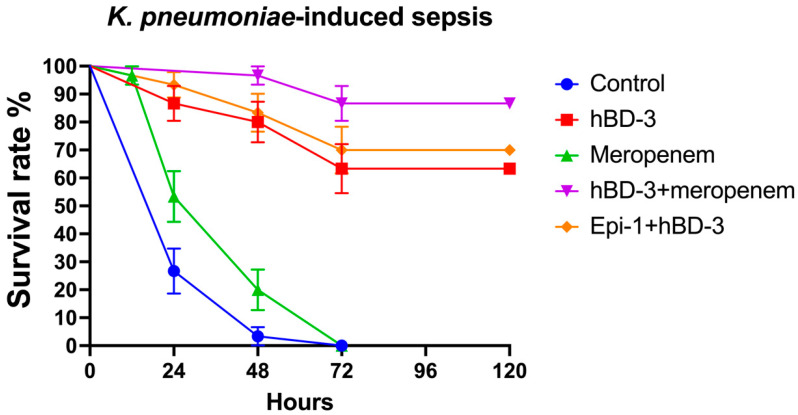
Survival rate in the *K. pneumoniae*-sepsis (CRKP_1 isolate was used)**.** hBD-3 (10 mg/kg), Epi-1 (10 mg/kg), and meropenem (25 mg/kg) were administered once, 30 min after infection (intraperitoneally). The control group was injected with sterile saline. Each group consisted of thirty mice. Bars show standard error of the mean.

**Figure 3 antibiotics-11-00076-f003:**
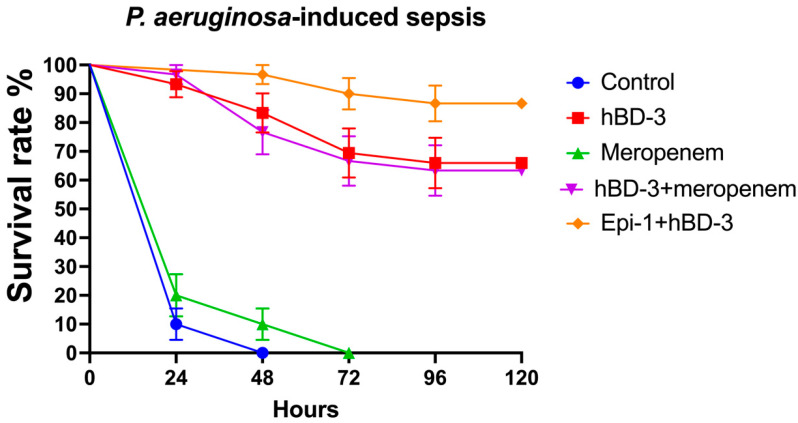
Survival rate in the *P. aeruginosa*-sepsis (CRPA_3 isolate was used). hBD-3 (10 mg/kg), Epi-1 (10 mg/kg), and meropenem (25 mg/kg) were administered once, 30 min after infection (intraperitoneally). The control group was injected with sterile saline. Each group consisted of thirty mice. Bars show standard error of the mean.

**Table 1 antibiotics-11-00076-t001:** Amino acid sequences and characteristics of AMPs used.

Peptide	Amino Acid Sequence	Length	Molecular Weight	Charge	Hydrophobic Residues
hBD-3	GIINTLQKYYCRVRGGRCAVLSCLPKEEQIGKCSTRGRKCCRRKK	45	5.17 kDa	+11	33%
Epi-1_22–42_	GFIFHIIKGLFHAGKMIHGLV	21	2.34 kDa	+3	57%

**Table 2 antibiotics-11-00076-t002:** Minimum inhibitory concentration (MIC, presented as mg/L and µM, in brackets) and fractional inhibitory concentration index (FICI) for methicillin-resistant *S. aureus* isolates (MRSA; *n* = 22).

Bacterial Isolates	Epi-1 MIC	Vancomycin MIC	FICI
MRSA_1	16 (6.8)	1 (0.69)	1.25
MRSA_2	16 (6.8)	2 (1.38)	0.5 *
MRSA_3	32 (13.7)	2 (1.38)	0.375 *
MRSA_4	8 (3.4)	2 (1.38)	0.75
MRSA_5	16 (6.8)	2 (1.38)	0.5 *
MRSA_6	8 (3.4)	1 (0.69)	1.125
MRSA_7	8 (3.4)	2 (1.38)	0.75
MRSA_8	16 (6.8)	2 (1.38)	0.5 *
MRSA_9	32 (13.7)	2 (1.38)	0.5 *
MRSA_10	16 (6.8)	2 (1.38)	0.75
MRSA_11	16 (6.8)	4 (2.76)	0.75
MRSA_12	16 (6.8)	2 (1.38)	0.5 *
MRSA_13	16 (6.8)	2 (1.38)	1
MRSA_14	16 (6.8)	2 (1.38)	0.75
MRSA_15	16 (6.8)	2 (1.38)	0.5 *
MRSA_16	16 (6.8)	4 (2.76)	0.3125 *
MRSA_17	8 (3.4)	4 (2.76)	0.5 *
MRSA_18	16 (6.8)	4 (2.76)	0.75
MRSA_19	16 (6.8)	1 (0.69)	0.75
MRSA_20	16 (6.8)	2 (1.38)	1
MRSA_21	16 (6.8)	2 (1.38)	0.5 *
MRSA_22	8 (3.4)	1 (0.69)	0.75

* Denotes synergistic effect.

**Table 3 antibiotics-11-00076-t003:** Minimum inhibitory concentration (MIC, presented as mg/L and µM, in brackets) and fractional inhibitory concentration index (FICI) for carbapenem-resistant *K. pneumoniae* (CRKP; *n* = 23) isolates.

Bacterial Isolates	Epi-1 MIC	hBD-3 MIC	FICI
CRKP_1	8 (3.4)	4 (0.8)	1
CRKP_2	16 (6.8)	4 (0.8)	0.75
CRKP_3	8 (3.4)	4 (0.8)	0.5 *
CRKP_4	8 (3.4)	8 (1.6)	0.375 *
CRKP_5	8 (3.4)	4 (0.8)	0.5 *
CRKP_6	16 (6.8)	2 (0.4)	1.25
CRKP_7	8 (3.4)	4 (0.8)	0.5 *
CRKP_8	8 (3.4)	4 (0.8)	1
CRKP_9	8 (3.4)	4 (0.8)	1
CRKP_10	4 (1.7)	8 (1.6)	1
CRKP_11	4 (1.7)	8 (1.6)	1.25
CRKP_12	16 (6.8)	16 (3.1)	1.5
CRKP_13	16 (6.8)	8 (1.6)	1.5
CRKP_14	8 (3.4)	8 (1.6)	0.375 *
CRKP_15	4 (1.7)	8 (1.6)	0.625
CRKP_16	8 (3.4)	2 (0.4)	1
CRKP_17	8 (3.4)	4 (0.8)	1.5
CRKP_18	8 (3.4)	8 (1.6)	0.75
CRKP_19	8 (3.4)	8 (1.6)	1.5
CRKP_20	8 (3.4)	8 (1.6)	1
CRKP_21	8 (3.4)	4 (0.8)	0.75
CRKP_22	16 (6.8)	4 (0.8)	0.75
CRKP_23	16 (6.8)	4 (0.8)	1

* Denotes synergistic effect.

**Table 4 antibiotics-11-00076-t004:** Minimum inhibitory concentration (MIC, presented as mg/L and µM, in brackets) and fractional inhibitory concentration index (FICI) for carbapenem-resistant *K. aerogenes* (CRKA; *n* = 17) isolates.

Bacterial Isolates	Epi-1 MIC	hBD-3 MIC	FICI
CRKA_1	16 (6.8)	2 (0.4)	2
CRKA_2	8 (3.4)	4 (0.8)	0.75
CRKA_3	4 (1.7)	2 (0.4)	0.75
CRKA_4	16 (6.8)	4 (0.8)	0.5 *
CRKA_5	4 (1.7)	4 (0.8)	0.75
CRKA_6	16 (6.8)	2 (0.4)	0.75
CRKA_7	4 (1.7)	2 (0.4)	0.75
CRKA_8	8 (3.4)	2 (0.4)	0.75
CRKA_9	16 (6.8)	2 (0.4)	1
CRKA_10	8 (3.4)	4 (0.8)	0.5 *
CRKA_11	4 (1.7)	4 (0.8)	0.5 *
CRKA_12	8 (3.4)	2 (0.4)	0.75
CRKA_13	16 (6.8)	4 (0.8)	0.375 *
CRKA_14	8 (3.4)	4 (0.8)	0.75
CRKA_15	8 (3.4)	2 (0.4)	0.625
CRKA_16	8 (3.4)	4 (0.8)	0.375 *
CRKA_17	4 (1.7)	4 (0.8)	0.75

* Denotes synergistic effect.

**Table 5 antibiotics-11-00076-t005:** Minimum inhibitory concentration (MIC, presented as mg/L and µM, in brackets) and fractional inhibitory concentration index (FICI) for carbapenem-resistant *P. aeruginosa* (CRPA; *n* = 13) isolates.

Bacterial Isolates	Epi-1 MIC	hBD-3 MIC	FICI
CRPA_1	8 (3.4)	4 (0.8)	0.5 *
CRPA_2	16 (6.8)	8 (1.6)	0.5 *
CRPA_3	8 (3.4)	4 (0.8)	1
CRPA_4	16 (6.8)	4 (0.8)	1
CRPA_5	16 (6.8)	16 (3.1)	0.5 *
CRPA_6	8 (3.4)	8 (1.6)	0.75
CRPA_7	4 (1.7)	4 (0.8)	0.75
CRPA_8	16 (6.8)	2 (0.4)	1.5
CRPA_9	8 (3.4)	2 (0.4)	1.5
CRPA_10	16 (6.8)	4 (0.8)	1.5
CRPA_11	16 (6.8)	8 (1.6)	0.5 *
CRPA_12	16 (6.8)	8 (1.6)	0.75
CRPA_13	32 (13.7)	16 (3.1)	0.5 *

* Denotes synergistic effect.

**Table 6 antibiotics-11-00076-t006:** Minimum inhibitory concentration (MIC, presented as mg/L and µM, in brackets) and fractional inhibitory concentration index (FICI) for carbapenem-resistant *A. baumannii* (CRAB; *n* = 9) isolates.

Bacterial Isolates	Epi-1 MIC	hBD-3 MIC	FICI
CRAB_1	32 (13.7)	16 (3.1)	0.5 *
CRAB_2	32 (13.7)	8 (1.6)	0.5 *
CRAB_3	4 (1.7)	8 (1.6)	0.5 *
CRAB_4	16 (6.8)	4 (0.8)	0.75
CRAB_5	32 (13.7)	4 (0.8)	1.25
CRAB_6	16 (6.8)	4 (0.8)	0.75
CRAB_7	32 (13.7)	8 (1.6)	0.5 *
CRAB_8	8 (3.4)	16 (3.1)	0.5 *
CRAB_9	16 (6.8)	4 (0.8)	0.75

* Denotes synergistic effect.

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
