# Peer review of "Antimicrobial Peptides Epinecidin-1 and Beta-Defesin-3 Are Effective against a Broad Spectrum of Antibiotic-Resistant Bacterial Isolates and Increase Survival Rate in Experimental Sepsis"

_antibiotics, 2022, doi:10.3390/antibiotics11010076_

Round 1

Reviewer 1 Report

General comment

In the manuscript, the antimicrobial activities of Epi-1 + vancomycin and Epi-1 + hBD-3 against methicillin- and carbapenem-resistant isolates were investigated, and the peptides showed antibacterial activity against all studied 16 clinical isolates. In experimental models of murine sepsis, hBD-3 monotherapy, as well as combinations of hBD-3 + meropenem and hBD-3 + Epi-1, increased survival rates. The results suggest that hBD-3 and Epi-1 are the promising candidates for multidrug-resistant infections caused by carbapenem-resistant bacteria.

Major revision

  • Figure 1: It is recommended to show the disulfide bonds of human beta-defensin-3 in Figure 1a.
  • Table 1: It is recommended to revise Ept-1 to Epi-122-42 according to Fig. 2 of Ref.11.
  • Table 2~6: As the molecular weight of Epi-1, vancomycin and hBD-3 was different each other, it is recommended to add MIC values as µM, in addition to mg/L.

Author Response

Thanks a lot for fair recommendations. I fixed everything.

Point 1. Figure 1: It is recommended to show the disulfide bonds of human beta-defensin-3 in Figure 1a. 

Response 1. I've added disulfide bonds of human beta-defensin-3 in Figure 1a. I've also a little bit changed the sizes if both Figures 1a and 1b - just to make their sizes similar to each other. 

Point 2. Table 1: It is recommended to revise Epi-1 to Epi-122-42 according to Fig. 2 of Ref.11.

Response 2. I've revised Epi-1 to Epi-122-42. 

Point 3. Table 2~6: As the molecular weight of Epi-1, vancomycin and hBD-3 was different each other, it is recommended to add MIC values as µM, in addition to mg/L.

Response 3. I've added MIC values in Tables 2, 3, 4, 5 and 6 as µM. 

Reviewer 2 Report

In this work A. Bolatchiev analyzed antimicrobial properties of two peptides: Epinecidin-1 and Beta-defensin-3 both in vitro and in vivo. Due to the widespread antibiotic resistance, topic is of huge importance, as Author mentioned in Introduction. Overall manuscript is very well written, results are promising, especially that analyses were performed on clinical isolates. Some specific comments below:

Line 28, 29: please provide full name of CDCs, US

Line 35: U.S. please standardize (once Author use US, once United States or U.S).

Line 43: Can Author explain what means several membranes?

Line 62: Gram with capital letter (please check whole manuscript).

Please fit Figure 1 to the page, it does not need to be that big.

Line 90: Please provide full name of MIC and FICI (it is provided in the description of Tables 2-6, while it suppose to be provided earlier in the manuscript).

Figure 2, 3: K. pneumoniae, P. aeruginosa in italics. Is it possible to include standard deviations maintaining clarity of figure? Maybe Author should include table as a supplementary material with detailed information?

Author Response

Thanks a lot for fair recommendations. I fixed everything. 

Point 1. Line 28, 29: please provide full name of CDCs, US

Response 1: I've provided full name of CDCs, US

Point 2. Line 35: U.S. please standardize (once Author use US, once United States or U.S).

Response 2: I'v standartized – used US... 

Point 3. Line 43: Can Author explain what means several membranes?

Response 3: This was my mistake and typo. I have removed this word from the text.

Point 4. Line 62: Gram with capital letter (please check whole manuscript).

Response 4: I've changed the word "Gram" with capital letter in whole text. 

Point 5. Please fit Figure 1 to the page, it does not need to be that big.

Response 5: I've added disulfide bonds (according to recommendation of Reviewer 1) and cropped the Figure 1 – made it smaller. 

Point 6. Line 90: Please provide full name of MIC and FICI (it is provided in the description of Tables 2-6, while it suppose to be provided earlier in the manuscript).

Response 6: I've provided the full names of MIC and FICI. 

Point 7. Figure 2, 3: K. pneumoniae, P. aeruginosa in italics. Is it possible to include standard deviations maintaining clarity of figure? Maybe Author should include table as a supplementary material with detailed information?

Response 7: I've changed to italics style the names of bacteria and included standard errors of the mean in figures 2 and 3. Also I included Tables in Raw Data – where there is survival analysis details with tables. 

If it is need more additional information, I'll be glad to provide it.